# The Partial Removal of Rectus Abdominis Muscle Inserting into Ribs in Ipsilateral Pedicled TRAM Flap for Breast Reconstruction

**DOI:** 10.3390/jcm11226647

**Published:** 2022-11-09

**Authors:** Daegu Son, Jaehoon Kim

**Affiliations:** Department of Plastic and Reconstructive Surgery, Keimyung University School of Medicine, Daegu 42601, Korea

**Keywords:** breast reconstruction, pedicled TRAM flap, muscle sparing, surgical technique

## Abstract

Background: The purpose of this study was to introduce a new surgical technique for pedicled TRAM flap that removes a part of the rectus abdominis muscle inserting into ribs, and to analyze this technique in comparison with classical pedicled TRAM flap. Methods: A retrospective review of patient charts from May 2006 to February 2016 was performed. The patient group that underwent the removal of the part of the rectus abdominis that inserts into the thorax (partial muscle resection; PMR group) was compared with the group that did not undergo this muscle resection (Classical group). The complications and aesthetic effects of surgery between the two groups were analyzed. Results: There were 34 patients in the classical group and 28 in the PMR group. There were no significant differences in postoperative complications between these two groups. The rates of fat necrosis were 32.1% in the PMR group and 36.1% in the classical group. The postoperative aesthetic outcome of the inframammary fold showed no significant differences in outcome between the classical and PMR groups. However, all items received higher scores in the PMR group. Conclusions: The authors’ new surgical method was associated with a positive cosmetic effect of improving inframammary fold aesthetics and could thus represent a new option for pTRAM breast reconstruction.

## 1. Introduction

The pedicled transverse rectus abdominis myocutaneous flap (pTRAM) technique, developed in 1982, is an innovative method of reconstructive breast surgery that hides the transverse scar via concomitant abdominoplasty [1]. However, since the development of free TRAM (fTRAM), which uses the inferior epigastric artery as a pedicle, pTRAM using the superior epigastric artery with a relatively low blood flow has become less popular [2,3,4].

With the development of reconstructive microsurgery, breast reconstruction surgery has surpassed fTRAM with the use of the deep inferior epigastric artery perforator (DIEP) flap technique. Despite its early promise, pTRAM has not advanced much. Although several surgical refinements, such as vascular delay [5,6], bilateral pedicle usage [7,8], partial rectus muscle preservation [9,10], and vascular augmentation [11] were developed in the late 1990s, in recent years, there has been no further progress in surgical techniques, and surgeons consider pTRAM the next best option behind fTRAM or DIEP flap breast reconstruction.

Reports of abdominal bulging or hernia due to the weakening of the abdominal wall, which is the most worrying complication of pTRAM breast reconstruction surgery, are variable and controversial [12,13,14,15]. According to a recent meta-analysis [16] and evidence-based clinical practice guidelines [17], the complications and results of pTRAM are no different from those of fTRAM or DIEP breast reconstruction. Thus, pTRAM needs to be further refined and developed. If microsurgery is not feasible for any reason, pTRAM is still a very important option for breast reconstruction surgery.

The authors of the present study have developed a new surgical technique that improves upon the shortcomings of the conventional pTRAM flap, in which the epigastric bulging and inframammary fold become unclear. The purpose of this study was to introduce a new surgical technique of pedicled TRAM flap that removes a part of the rectus abdominis muscle inserting into the ribs and to analyze this technique with conventional pedicled TRAM flap.

## 2. Materials and Methods

Sixty-two consecutive patients who underwent breast reconstruction surgery with ipsilateral pedicled TRAM from 2006 to 2016 were studied. Of these, 34 patients underwent conventional pTRAM (Classical pTRAM) surgery, and 28 patients underwent a partial resection of the muscle at the site where the rectus abdominis attaches to the ribs (partial muscle resection; PMR pTRAM) (Table 1). At the beginning of the pTRAM breast reconstruction, conventional pTRAM was mainly used. However, after developing a new surgical method by analyzing the anatomy and blood circulation of the rectus abdominis muscle through cadaveric dissection, PMR pTRAM was mainly performed. In some cases, a part of the rectus abdominis muscle was left and preserved, but this was not used as a classification criterion for patients. All patients with delayed breast reconstruction underwent total mastectomy, and most patients with immediate reconstruction underwent skin-sparing mastectomy. Seven patients in the classical pTRAM group underwent nipple-sparing mastectomy.

The study was approved by the appropriate ethics review boards at our institution (2017-08-039).

### 2.1. Chart Review

Retrospective review of patient charts from May 2006 to February 2016 was performed. Demographic data, body mass index, incidence of delayed reconstruction, incidence of vertical scar in the lower abdomen, hypertension, contralateral breast augmentation, reduction, mastopexy, and preoperative and postoperative radiation were assessed.

Complications at reconstructed breast and donor sites were investigated. Complications in reconstructed breasts included pTRAM flap necrosis, skin necrosis of skin-sparing mastectomy, hematoma, seroma, wound dehiscence, and fat necrosis. Fat necrosis was defined as a palpable firm mass with a diameter of 1 cm or more. Complications at the donor site included hematoma, seroma, abdominal hernia, abdominal bulge, and wound dehiscence.

### 2.2. Surgical Technique

#### 2.2.1. Preoperative Design and Incision

The surface line of the ipsilateral rectus abdominis muscle was identified while the patient held their breath and strengthened the muscle in an upright position. The upper incision was placed just above the upper border of the umbilicus, and an additional dotted line was drawn 4–5 cm above this line. The amount of excision in zones 3 and 4 was determined according to the vertical and horizontal dimensions and shape of the opposite breast. Usually, most of zone 4 was removed (Figure 1).

An incision was made to the dermis on the upper incision line with a number 10 blade and dissection proceeded obliquely upward to the dotted line with a Bovie, so the subcutaneous fat could be added into the pTRAM flap, and the lower border of the abdominal flap was thinned. After complete elevation of the upper abdominal flap, a tunnel was made from the xiphoid process to the medial third of the inframammary fold for transposition of the pTRAM flap from abdomen to breast.

#### 2.2.2. Flap Elevation

Elevation of the contralateral (non-pedicle-side) TRAM flap was performed up to 1 cm past the midline linea alba, and the pedicle side was elevated to the level of the perforators. The medial and lateral third of the ipsilateral anterior rectus sheath was separated from the muscle completely with unipolar and bipolar electrocautery with preservation of the central third of the anterior rectus sheath. In the muscle-sparing cases, the lateral third of the rectus abdominis muscle was split longitudinally, preserving 3 or 4 intercostal nerves innervating the muscle. The width of the conserved muscle was 2–3 cm (Figure 2).

In most cases, the intercostal nerve that innervates the rectus muscle is very close to the lateral row of the deep inferior artery [18]. Thus, most nerves have to be cut and replanted into the muscles. Under a microscope, the end of the cut nerve was inserted between the rectus muscle fibers that were open, and the epineurium of the nerve and the epimysium of the muscle were sutured with 9-0 nylon. Usually, 3 or 4 intercostal nerves located cephalic and caudal to the umbilicus, including a large one that innervates a spared rectus abdominis muscle at the level of the arcuate line, should be selected for neurotization (Figure 3 and Figure 4).

#### 2.2.3. Partial Removal of Rectus Abdominis Muscle Inserting into Ribs and Flap Insetting

A rectus abdominis muscle inserted into the costal cartilage of the fifth, sixth and seventh ribs was resected with sufficient width and length as to receive the flipped muscle pedicles for transposition. Muscle resection must be limited to just above and lateral to the costal margin to ensure muscle preservation and to avoid injury to the superior epigastric artery located deep inside and along the costal margin. After muscle resection, any bleeding points should be thoroughly cauterized, as there are usually 1–2 perforators in this area. The bottom of this tunnel consists of the ribs and cartilage; the medial limit was the xyphoid, and the lateral limit was the preserved muscle. The pTRAM pedicle was placed in the widened tunnel formed by muscle removal at the bottom (Figure 5 and Figure 6). After the pTRAM flap had been completely transferred, it was easy to see the deep superior epigastric artery (DSEA) enter the muscle. Further cutting of the rectus muscle to prevent the pTRAM pedicle from being compressed in the tunnel ensured improvement of blood circulation (Appendix A).

#### 2.2.4. Dorsite Closure and Breast Shaping

The anterior rectus sheaths were closed with 1-0 vicryl without any mesh. The transpositioned pTRAM is flipped over such that the cutaneous flap was on top. In this state, if the flaps were placed in parallel positions to the left and right, the umbilicus was located at the top center. If this position is maintained, the breast takes on a breast shape with a wide base from side to side (horizontal). If the pTRAM is rotated 90 degrees laterally, the place where the umbilicus was located moves to the lateral side, and the tissue takes on a narrow breast shape that is more vertical. The shape of most breasts is somewhere between the two extremes (horizontal and vertical). The most common breast shape is slightly oblique to the lateral side. Since the pedicle length of the ipsilateral pTRAM is sufficient, it is possible to create a sufficiently symmetric breast with proper rotation, folding, and fixation (Figure 7).

### 2.3. Aesthetic Evaluation of the Inframammary Fold

In our surgical method, the primary purpose of removing part of the rectus abdominis muscle at the thorax is to reduce epigastric bulging, thereby probably reducing the pressure on the pedicle and improving the venous congestion secondarily. The postoperative aesthetic outcome of the inframammary fold (IMF) and epigastric bulging was evaluated by 3 plastic surgeons who did not participate in the surgery.

In photographs taken 1 year after breast reconstruction, (1) IMF continuity, (2) IMF definition, and (3) symmetry were evaluated on an ordinal scale. This analysis adopted Yoon’s research criteria [19], except that the evaluators used a scoring system ranging from 1 (lowest) to 5 (highest) points per item. The total score was divided into four categories: excellent (15–13), good (12–10 points), fair (9–7 points), and poor (less than 6 points). Regardless of the presence or absence of muscle sparing, this analysis compared the group in which the rectus abdominis muscle was preserved with the group in which it was partially removed from the ribs.

### 2.4. Statistical Analysis

Continuous data are expressed as median and inter quartile range (IQR), and categorical data are shown as frequency and percentage. In comparisons between the two groups, the Mann–Whitney U test was used for continuous data (mean difference), and the chi-square test was used for categorical data (proportional difference); for the latter, if more than 20% of cells had an expected frequency of less than 5, Fisher’s exact test was performed. Statistical analysis was performed using SAS 9.4 (Copyright (c) 2002–2012 by SAS Institute Inc., Cary, NC, USA). The significance level was set to 5%.

## 3. Results

During the study period, 34 patients underwent conventional pTRAM surgery (classical pTRAM group), whereas 28 patients underwent partial muscle resection pTRAM surgery (PMR pTRAM group). The average ages of the classical and PMR groups were 46.2 and 43.9 years old, respectively, with average BMIs of 23.8 and 22.9 kg/m^2^, respectively. None of the subjects were smokers. Immediate reconstructive surgery was conducted in 30 (88.2%) cases in the classical group and 17 (60.7%) cases in the PMR group. The average follow-up periods for the classical and PMR groups were 18.8 and 19.6 months, respectively (Table 1).

The postoperative complications reported in the classical group were pTRAM flap skin necrosis in 3 (8.8%) patients, mastectomy skin necrosis in 3 (8.8%) patients, fat necrosis in 12 (35.3%) patients, and wound dehiscence in 2 (5.9%) patients. There was also 1 (2.9%) patient with abdominal hernia and 1 (2.9%) patient with abdominal skin necrosis due to donor site complications. All wounds healed with conservative treatment. The postoperative complications in the PMR group were pTRAM flap skin necrosis in 1 (3.6%) patient, mastectomy skin necrosis in 3 (3.6%) patients, wound dehiscence in 2 (7.1%) patients, and fat necrosis in 9 (32.1%) patients. There was also 1 (3.6%) patient with abdominal hernia and 1 (3.6%) patient with abdominal skin necrosis due to donor site complications (Table 2). There were no significant differences in the rates of postoperative complications between the two groups.

Two patients in the classical group and one patient in the PMR group with breast skin necrosis were healed via skin graft. The remaining complications were cured with conservative treatment. Necrotic fat was surgically removed under local anesthesia in both groups. Two patients with lower abdominal bulging underwent correction using a polypropylene mesh (Prolene^®^, Johnson & Johnson, New Brunswick, NJ, USA), while the other was treated with a simple plication with 1-0 and 2-0 Monosyn suture materials (B. Brown, Barcelona, Spain).

There were no significant differences in fat necrosis between the classical and PMR groups. In the classical group, the fat necrosis rate was 26.7% in immediate reconstruction and 100% in delayed reconstruction patients. In the PMR group, the fat necrosis rate was 29.4% in immediate reconstruction and 36.4% in delayed reconstruction patients (Table 3).

The post-operative aesthetic outcome of the IMF was evaluated as good in terms of continuity, definition, and symmetry, and there was no significant difference between the partial muscle resection group (Figure 8 and Figure 9) and the entire muscle group (Figure 10). However, in the partial muscle resection group, all items were rated better (Table 4).

## 4. Discussion

In the early stages of the development of the technique, pTRAM flap breast reconstruction used the contralateral side as a pedicle, but problems arose such as increased tension in the pedicle resulting in venous congestion and disappearing inner IMF due to perixiphoid bulge across the midline [20,21]. Currently, the ipsilateral pedicle is preferred because it has many advantages. Its use eliminates the perixiphoid bulge, produces a longer pedicle allowing for greater mobility, creates a tunnel at the center of the inframammary fold by preserving the medial attachment of the inframammary crease ligament, grants versality in breast mound shaping, facilitates flap insetting, and results in good venous drainage [22,23,24,25].

In addition to the known advantages of the ipsilateral pedicle, the author found that cosmetically better results could be obtained by removing a part of the rectus muscle inserted into the thorax. The pressure or resistance on the pedicle by the tunnel was expected to be reduced. For this reason, the venous drainage of the flap itself is not disturbed internally and epigastric bulging is not prominent externally. If the operator only wants to improve venous congestion, the solution is to widen the tunnel. However, it is difficult to obtain good cosmetic results through this process because the area where the IMF is lost will become wider. Therefore, the PMR pTRAM technique is a good way to widen the tunnel vertically rather than horizontally. In terms of preserving the medial 1/3 of the rectus abdominis muscle in the abdomen, this effect might be greater because the volume of the pedicle has been reduced by that much compared with taking the whole muscle. Although this study did not yield statistically significant results, cosmetically better results were obtained in the PMR pTRAM group than the classical pTRAM group (Table 4). The author’s surgery is expected to be useful in patients with relatively small breasts and clearly visible inframammary folds or those with a protruding rib cage (Figure 8).

Although the PMR pTRAM technique may initially appear dangerous because it is performed near the pedicle entering the muscle, it is very safe. The site where the superior epigastric artery enters the posterior surface of the rectus abdominis passes through the costal margin, so the muscle can be safely removed from the superior of the costal margin (Figure 5 and Figure 6) (Appendix A).

According to the current results, the frequencies of fat necrosis are both somewhat high at 36.1% in the classical pTRAM group and 32.1% in the PMR pTRAM group, and partial flap necrosis is present, but most cases are mild and can easily be corrected under local anesthesia immediately after the necrotic area is established or during a secondary touch-up procedure. The higher fat necrosis rate in our patients with delayed reconstruction vs. immediate reconstruction may be due to the inability to completely excise zone 2, in which the blood supply is poor because a large amount of skin and soft tissue is needed for delayed breast reconstruction. Even in immediate breast reconstruction patients, fat necrosis occurred mostly in zones 2 and 4. Accordingly, the authors posit that the increase in fat necrosis was caused by the insufficient resection of zone 2. Therefore, the most reliable way to reduce fat necrosis in pTRAM may be to completely remove areas with poor blood supply in advance. In the report of Yoon et al.’s island-type pTRAM breast reconstruction, the rate of fat necrosis was very low (9%) because half of zone 2 was removed [19]. The use of angiography with indocyanine green fluorescent dye may help to reliably determine the degree of perfusion [26]. In order to fully utilize the advantages of pTRAM and minimize fat necrosis, the surgeon should aim to completely remove the area in which perfusion decreases in zone 2. Therefore, pTRAM is the most suitable option for patients with small to moderate size breasts (less than a large B cup) and no midline vertical scars in immediate reconstruction.

The partial preservation of the rectus abdominis without the intercostal nerve is useless because the denervated muscle will atrophy [27]. The innervation of the rectus muscle is segmental from the 7th to 12th intercostal nerve, with most nerves entering the muscle in the lateral third in direct association with the lateral intercostal vascular pedicle, though two to three nerves enter the lower portion of the muscle below the umbilicus lateral to the inferior epigastric artery [18]. For that reason, it is sometimes not possible to preserve the intercostal nerve while preserving the lateral rectus abdominis (Figure 4). In such cases, as shown in our previous report, neurotization is an effective method of preventing muscle atrophy [28]. The preservation of the neurotized lateral third of the rectus muscle resulted in no differences in the rates of surgical complications or fat necrosis. This means that leaving part of the muscle does not interfere with blood circulation in the flap, nor does it affect donor site morbidity. However, in order to spare the rectus muscle, the nerve must be preserved, and if the nerve is too close to the pedicle, preservation is difficult, so neurotization must be performed.

This study has certain limitations. First, this study was a retrospective study, and there were inevitably limitations in statistical power due to the insufficient number of patients. Another limitation was in the statistical analysis of cosmetic excellence. Although the contour of the IMF is affected by various factors, such as the shape of the patient’s chest, the size of the breast, and the degree of ptosis of the breast, the authors compared them in a simple manner without considering these factors. However, the author’s new surgical technique is expected to inspire surgeons who perform breast reconstruction surgery with pedicled TRAM flap and be applied to various techniques.

## 5. Conclusions

The authors’ new surgical method suggests a method that preserves the contours of the IMF as much as possible to obtain better cosmetic results and which can be safely performed. In the case of preserving a part of the rectus abdominis muscle in pTRAM breast reconstruction surgery, the nerves must be preserved, and, if necessary, neurotization is recommended.

## Figures and Tables

**Figure 1 jcm-11-06647-f001:**
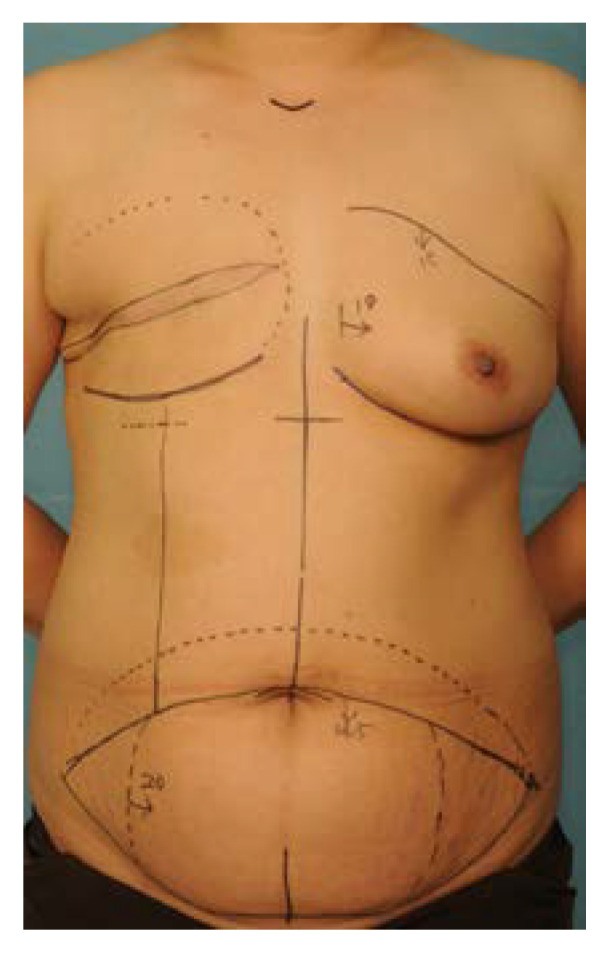
Design of ipsilateral pedicled TRAM flap for delayed breast reconstruction. On the breast side, the mastectomy scar was removed, the area to be dissected was marked with a dotted line, and the IMF line was drawn about 2 cm above the opposite side. On the abdomen, the upper incision was placed just above the upper border of the umbilicus, and an additional dotted line was drawn 4–5 cm above this line.

**Figure 2 jcm-11-06647-f002:**
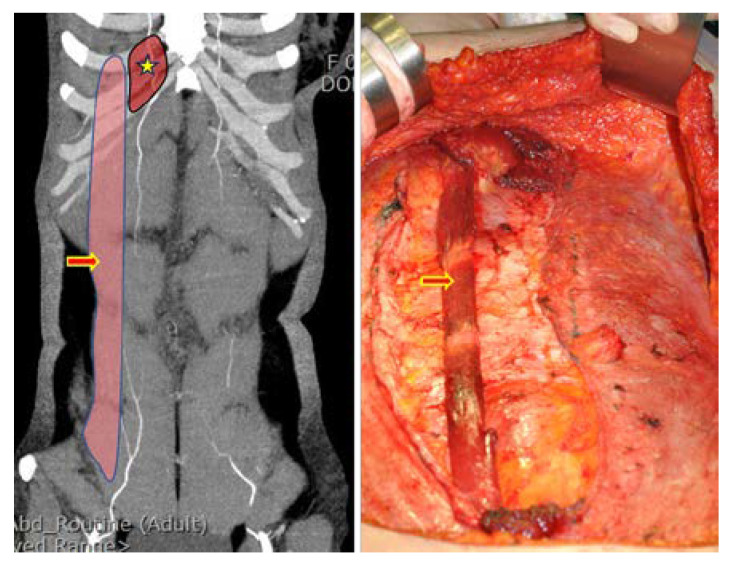
Preservation of the rectus abdominis muscle. (**Left**) Muscles can be preserved in the area marked light red and indicated by the arrow. The rectus muscle is removed from the red area indicated by the star. (**Right**) The width of the preserved rectus muscle is about 2–3 cm.

**Figure 3 jcm-11-06647-f003:**
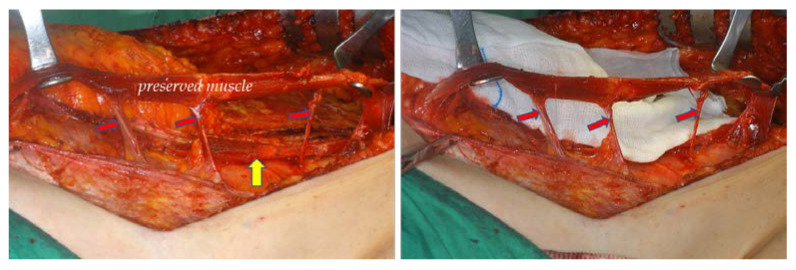
Preserved rectus muscle with innervated intercostal nerve (red arrows). The yellow arrow indicates the rectus muscle that is used as the pedicle, and the muscle held by the Army-Navy retractor is the preserved rectus muscle.

**Figure 4 jcm-11-06647-f004:**
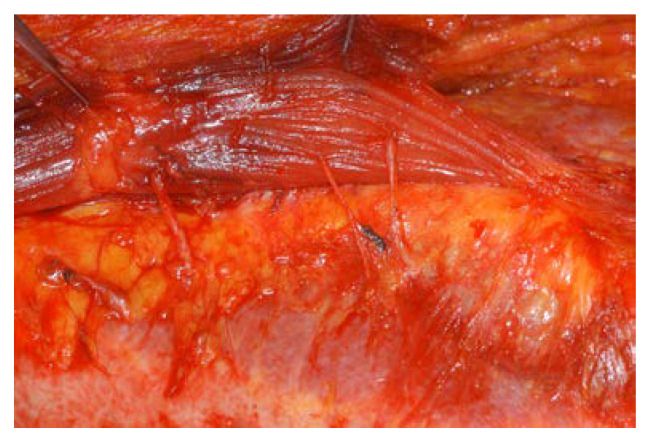
Neurotization of the cut intercostal nerves. Under the microscope, the end of the cut nerve was inserted between the rectus muscle fibers that were open, and the epineurium of the nerve and the epimysium of the muscle were sutured.

**Figure 5 jcm-11-06647-f005:**
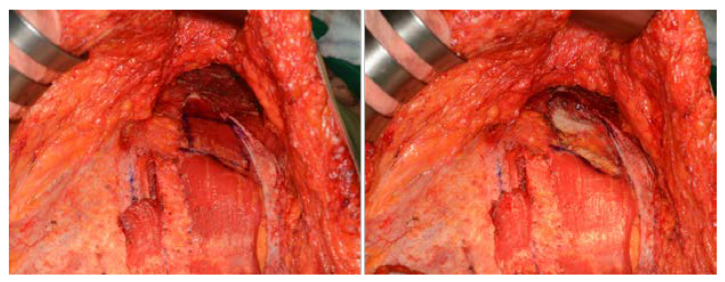
(**Left**) A rectus abdominis muscle inserted into the costal cartilage of the fifth, sixth, and seventh ribs was marked. (**Right**) Image taken after muscle removal. Muscle resection must be limited to just above and lateral to the costal margin to ensure muscle preservation.

**Figure 6 jcm-11-06647-f006:**
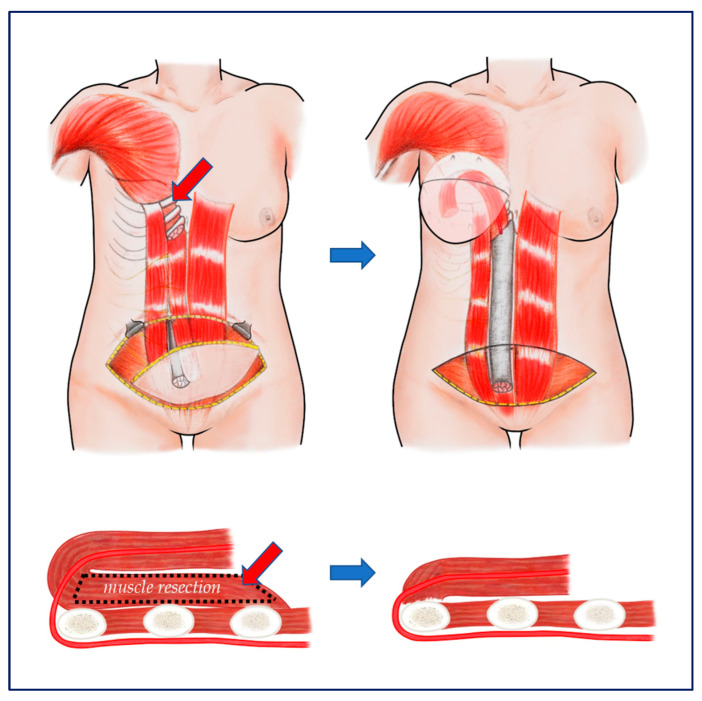
Schematic drawing of rectus muscle resection and flap transposition. (**Top left**) The area from which the rectus muscle was removed is marked by the red arrow. (**Top right**) After transposition of the pedicled TRAM flaps. (**Bottom**) A cross ssectional view of the area where the rectus muscle was transpositioned, the left side was when the muscle was not excised and the right side was when the muscle (red arrow) was resected. The height decreases as much as the width of the resected muscle, so the pedicle is subject to less pressure and epigastric bulging is avoided.

**Figure 7 jcm-11-06647-f007:**
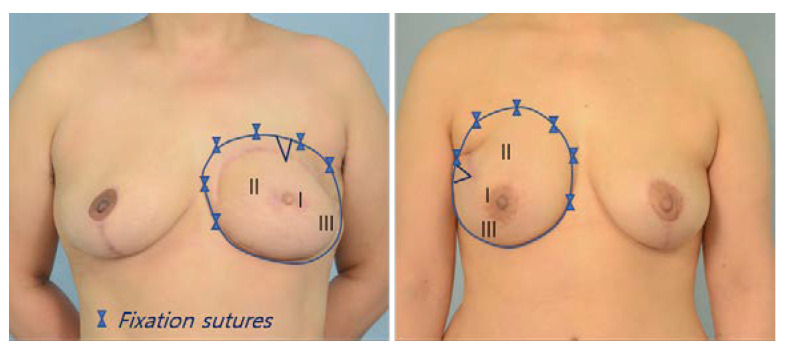
Breast shaping. Since the length of the pedicle is sufficient, the flap can be properly rotated according to the shape of the breast on the opposite side to create symmetry. The part marked with a V is the previous location of the belly button. Zone 2 is located in the medial and upper medial area of the horizontal breast (**Left**) and the upper and upper medial area of the longitudinal breast (**Right**); I, Zone 1; II, Zone 2; III, Zone 3.

**Figure 8 jcm-11-06647-f008:**
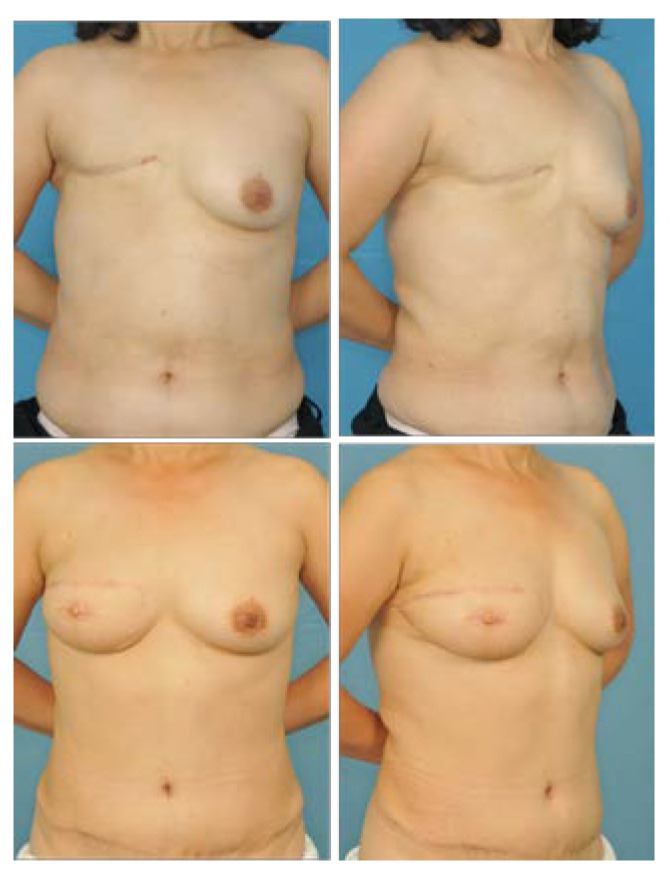
A 50-year-old patient who underwent delayed reconstruction with partial rectus muscle resection and innervated muscle sparing. (**Top**) Preoperative frontal and three-quarters oblique view. (**Bottom**) Frontal and three-quarters oblique view 28 months postoperative. In the aesthetic evaluation of the IMF, continuity, definition, and symmetry were given scores of 4, 4.3, and 4.3, respectively.

**Figure 9 jcm-11-06647-f009:**
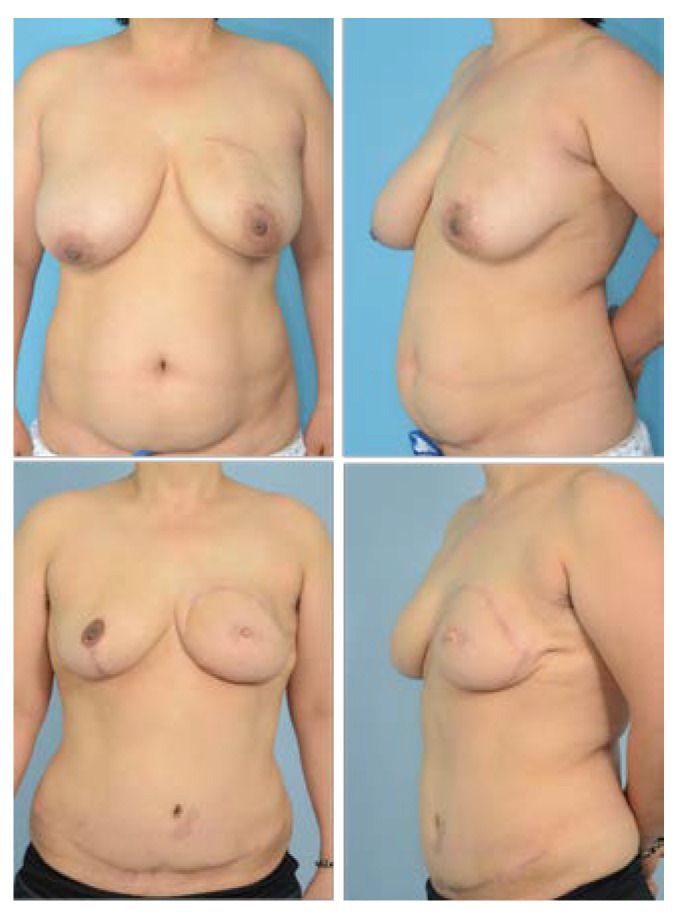
A 39-year-old patient who underwent immediate reconstruction of the left breast with partial muscle resection and innervated muscle sparing and breast reduction of the right breast. (**Top**) Preoperative frontal and three-quarters oblique view. (**Bottom**) Frontal and three-quarters oblique view 11 months post-operative. Fat necrosis in zone 2 was excised under local anesthesia. In the aesthetic evaluation of the IMF, continuity, definition, and symmetry were given scores of 3.7, 4, and 4, respectively.

**Figure 10 jcm-11-06647-f010:**
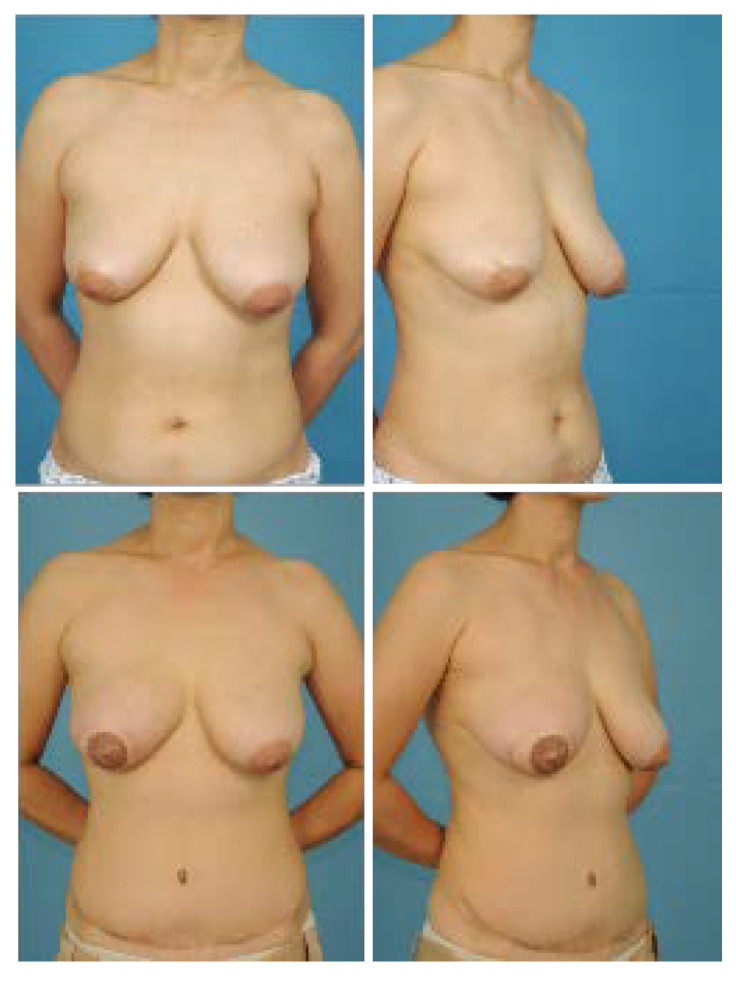
A 54-year-old patient who underwent immediate reconstruction of the right breast using the entire rectus muscle. (**Top**) Preoperative frontal and three-quarters oblique view. (**Bottom**) Frontal and three-quarters oblique view 15 months post-operative. In the aesthetic evaluation of the IMF, continuity, definition, and symmetry were given scores of 3.7, 4.3, and 4, respectively.

**Table 1 jcm-11-06647-t001:** Patient demographics.

	Classical pTRAM Group (%)	PMR pTRAM Group (%)	*p*-Value
No. of patients Median age (IQR), yr Mean body mass index (IQR), kg/m^2^	34 47.5 (7.75) 23.4 (3.73)	28 44.0 (12.0) 22.4 (3.83)	0.248 * 0.249 *
Breast reconstruction Immediate Delayed	30 (88.2) 4 (11.8)	17 (60.7) 11 (39.3)	0.012 ^†^ 0.012 ^†^
Vertical scar in the lower abdomen	1 (2.9)	4 (14.3)	0.166 ^‡^
Contralateral breast Augmentation Reduction Mastopexy	0 6 (17.6) 1 (2.9)	2 (7.1) 1 (3.6) 2 (7.1)	0.200 ^‡^ 0.116 ^‡^ 0.585 ^‡^
Radiation Preoperative Postoperative	3 (8.8) 5 (14.7)	0 (0.0) 2 (7.1)	0.245 ^‡^ 0.442 ^‡^

pTRAM, pedicled transverse rectus abdominis myocutaneous flap; PMR, partial muscle resection; IQR, inter quartile range, * Mann-Whitney U test, ^†^ chi-squared test, ^‡^ Fisher’s exact test.

**Table 2 jcm-11-06647-t002:** Reconstructed breast and donor site complications.

Complications	Classical pTRAM Group No. (%)	PMR pTRAM Group No. (%)	*p*-Value
Reconstructed Breast			
pTRAM skin necrosis	3 (8.8)	1(3.6)	0.620 ^‡^
Mastectomy skin necrosis	3 (8.8)	1 (3.6)	0.620 ^‡^
Hematoma	0 (0.0)	1 (3.6)	0.452 ^‡^
Wound dehiscence	2 (5.9)	2 (7.1)	1.000 ^‡^
Fat necrosis	12 (35.3)	9 (32.1)	0.794 ^†^
Donor site			
Abdominal bulging	1 (2.9)	1 (3.6)	1.000 ^‡^
Skin necrosis	1 (2.9)	1 (3.6)	1.000 ^‡^

TRAM, pedicled transverse rectus abdominis myocutaneous flap; PMR, partial muscle resection. ^‡^ Fisher’s exact test, ^†^ chi-squared test.

**Table 3 jcm-11-06647-t003:** Fat necrosis rate and relationship between classical and partial muscle resection pTRAM flap.

	Classical pTRAM Group (%)	PMR pTRAM Group (%)	*p*-Value
No. of patients with fat necrosis	12/34 (36.1)	9/28 (32.1)	0.794 ^†^
No. of patients with fat necrosis Immediate reconstruction Delayed reconstruction	8/30 (26.7) 4/4 (100)	5/17 (29.4) 4/11 (36.4)	1.000 ^‡^ 0.077 ^‡^

pTRAM, pedicled transverse rectus abdominis myocutaneous flap; PMR, partial muscle resection. ^‡^ Fisher’s exact test, ^†^ chi-squared test.

**Table 4 jcm-11-06647-t004:** Aesthetic evaluation of the inframammary fold.

	Classical pTRAM Group (%) (*n* = 31)	PMR pTRAM Group (%) (*n* = 27)	*p*-Value *
Continuity	11.0 (3.5)	11.0 (3.0)	0.220
Definition	11.0 (5.0)	12.0 (4.0)	0.209
Symmetry	10.0 (3.0)	11.0 (4.0)	0.324

PMR, partial muscle resection; EM, entire muscle. * Mann-Whitney U test.

## Data Availability

Not applicable.

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
