# Peer review of "The Partial Removal of Rectus Abdominis Muscle Inserting into Ribs in Ipsilateral Pedicled TRAM Flap for Breast Reconstruction"

_jcm, 2022, doi:10.3390/jcm11226647_

Round 1

Reviewer 1 Report

Dear Authors:

The manuscript "Partial Removal of Rectus Abdominis Muscle Inserting into Ribs and Innervated Muscle Sparing Techniques in Ipsilateral Pedicled TRAM Flap for Breast Reconstruction" has demonstrated new surgical method was associated with a positive cosmetic effect of improving inframammary fold aesthetics. I have just a few suggestions.

1. Some references or background information are missing. In page 1, line 35-37:"With the development of reconstructive microsurgery, breast reconstruction surgery has surpassed fTRAM with use of the deep inferior epigastric artery perforator (DIEP) flap technique." Please add more background information about these surgery methods. For example, some articles have demonstrated DIEP technique. (please cite: 1. Patient Management Strategies in Perioperative, Intraoperative, and Postoperative Period in Breast Reconstruction With DIEP-Flap: Clinical Recommendations. Front Surg. 2022 Feb 15;9:729181. doi: 10.3389/fsurg.2022.729181. PMID: 35242802; PMCID: PMC8887567.

2. Robot-Assisted Minimally Invasive Breast Surgery: Recent Evidence with Comparative Clinical Outcomes. J Clin Med. 2022 Mar 25;11(7):1827. doi: 10.3390/jcm11071827. PMID: 35407434; PMCID: PMC8999956.)

2. The manuscript needs linguistic improvement.

Best,

Author Response

As a result of careful consideration of the excellent comments of reviewers, the author simplified the setting of the control group to clarify the purpose of the study and improve the reader's understanding. The author's new surgical method is the PMR pTRAM surgery, which removes a part of the rectus abdominis muscle that attaches to the ribcage. Therefore, this study was modified to compare the PMR pTRAM group with the conventional pTRAM (Classical pTRAM) that uses all muscles. Existing studies confuse readers because they subdivided groups including the case where the lateral 1/3 of the rectus abdominis muscle was preserved in the abdomen and the case where it was not, and there were difficulties in statistical analysis. The author made major revisions to the manuscript, especially the discussion section, and performed statistical analysis again.

Answer) As you pointed out, 2 references were added, and the manuscript was revised again by an English expert.

Reviewer 2 Report

I have read with great interest the manuscript entitled "Partial Removal of Rectus Abdominis Muscle Inserting into Ribs and Innervated Muscle Sparing Techniques in Ipsilateral Pedicled TRAM Flap for Breast Reconstruction". Below my comments to the authors

1. My first comment is more of a reflection regarding the choice of the technique. By reading the manuscript it is obvious that the authors possess the microsurgical skills to perform free perforator flaps. The patient cohort includes patient up to 2016. Do the authors still use the pTRAM technique or do they offer more DIEP flaps in their practice to not sacrifice the muscle at all. 

2. The introduction section is very well written, I would suggest to state in a clear way the aim of the study to make it easier for the reader to understand. For example primary aim is to compare the two techniques, eventual secondary aims. 

3. In the material and methods section the authors describe patient characteristics. Did all the patients undergo radical mastectomy? or did some patients have other type of mastectomies. Can the authors give more details on that? In case some skin sparing or nipple sparing mastectomies were performed how did the authors planned skin integration? Do they have a standardised algorithm like " 10.1002/micr.22269".

4. In the material and methods section the authors describe the surgical technique I a very thorough way. I would ask the authors to consider explaining the technique by adding some further subparagraphs regarding the flap raising, flap insetting and stress in each of them the modifications they added and any tips and tricks for the novice reader.

5.  Did the authors find any difference regarding operative time between the two techniques? Did they analyse it? Of course there are a lot of factors influencing operative time but this, if available, might be another variable that can help reader appreciate the superiority of one or the other technique. " 10.1002/micr.30203"

6. The authors describe complications by referring to spared breast skin necrosis, maybe a more appropriate term would be mastectomy skin necrosis.

Author Response

I have read with great interest the manuscript entitled "Partial Removal of Rectus Abdominis Muscle Inserting into Ribs and Innervated Muscle Sparing Techniques in Ipsilateral Pedicled TRAM Flap for Breast Reconstruction". Below my comments to the authors

As a result of careful consideration of the excellent comments of reviewers, the author simplified the setting of the control group to clarify the purpose of the study and improve the reader's understanding. The author's new surgical method is the PMR pTRAM surgery, which removes a part of the rectus abdominis muscle that attaches to the ribcage. Therefore, this study was modified to compare the PMR pTRAM group with the conventional pTRAM (Classical pTRAM) that uses all muscles. Existing studies confuse readers because they subdivided groups including the case where the lateral 1/3 of the rectus abdominis muscle was preserved in the abdomen and the case where it was not, and there were difficulties in statistical analysis. The author made major revisions to the manuscript, especially the discussion section, and performed statistical analysis again.

  1. My first comment is more of a reflection regarding the choice of the technique. By reading the manuscript it is obvious that the authors possess the microsurgical skills to perform free perforator flaps. The patient cohort includes patient up to 2016. Do the authors still use the pTRAM technique or do they offer more DIEP flaps in their practice to not sacrifice the muscle at all. 

Answer) Thanks for the good question. The author has performed many free flaps as a microsurgeon. However, in breast reconstruction surgery, pTRAM is still performed if necessary. In any case, if microsurgery is not possible, pTRAM is still considered as one of the most important methods in breast reconstruction surgery, and the technique needs to be further developed.

  1. The introduction section is very well written, I would suggest to state in a clear way the aim of the study to make it easier for the reader to understand. For example primary aim is to compare the two techniques, eventual secondary aims. 

Answer) As pointed out by the reviewer, the aim of the study was clearly described again.

  1. In the material and methods section the authors describe patient characteristics. Did all the patients undergo radical mastectomy? or did some patients have other type of mastectomies. Can the authors give more details on that? In case some skin sparing or nipple sparing mastectomies were performed how did the authors planned skin integration? Do they have a standardised algorithm like " 10.1002/micr.22269".

Answer) All patients with delayed breast reconstruction underwent total mastectomy, and most patients with immediate reconstruction underwent skin sparing mastectomy. Seven patients in the classical pTRAM group underwent nipple sparing mastectomy. Added this to the description. The preserved breast skin and nipple were used for reconstructive surgery as much as possible, and only the part with poor blood circulation in the margin was excised. The author did not use the standardized algorithm.

  1. In the material and methods section the authors describe the surgical technique I a very thorough way. I would ask the authors to consider explaining the technique by adding some further subparagraphs regarding the flap raising, flap insetting and stress in each of them the modifications they added and any tips and tricks for the novice reader.

Answer) Thank you to the reviewers for their good comments. As you pointed out, the title has been further subdivided according to the operation progress and modified to help the reader's understanding.

  1.  Did the authors find any difference regarding operative time between the two techniques? Did they analyse it? Of course there are a lot of factors influencing operative time but this, if available, might be another variable that can help reader appreciate the superiority of one or the other technique. " 10.1002/micr.30203"

Answer) Unfortunately, we did not analyze the operation time. However, the author's surgical method differs from the conventional method in the following two ways. The first is excision of the insertion part of the rectus abdominis muscle in the ribcage, and the second is to preserve and neurotize a part of the rectus abdominis muscle. The time taken to excise the muscle was 5-10 minutes, and the time to preserve the muscle and neurotization was about 30-60 minutes.

  1. The authors describe complications by referring to spared breast skin necrosis, maybe a more appropriate term would be mastectomy skin necrosis.

Answer) Thank you for your thoughtful comments. I have edited it as you pointed out.

Reviewer 3 Report

The authors suggest a modified technique for breast reconstruction using the pedicled TRAM flap with or without "innervated rectus abdominis sparing". The method is of value, despite that advances in the field have rendered it an "outsider". The paper suffers small numbers over a long period of time, and this needs to be clarified. Since consecutive patients are reported, then this suggests that the method is reserved for appropriate candidates or that the caseload of the unit is small. This needs to be defined. The statistics are inevitably based on small numbers and differences that may or may not be clinically meaningful cannot reach significance. The authors choose to describe continuous variables with mean and SD, which is suboptimal, given the small numbers. I would suggest median and iqr with appropriate tests for comparisons.

The discussion is too long and not tightly based on the results, but rather extrapolative. It needs to be adjusted. A clear, evident result is absent. 

The manuscript yields some interest and merit, but needs to be drastically revised.

Author Response

The authors suggest a modified technique for breast reconstruction using the pedicled TRAM flap with or without "innervated rectus abdominis sparing". The method is of value, despite that advances in the field have rendered it an "outsider". The paper suffers small numbers over a long period of time, and this needs to be clarified. Since consecutive patients are reported, then this suggests that the method is reserved for appropriate candidates or that the caseload of the unit is small. This needs to be defined. The statistics are inevitably based on small numbers and differences that may or may not be clinically meaningful cannot reach significance. The authors choose to describe continuous variables with mean and SD, which is suboptimal, given the small numbers. I would suggest median and iqr with appropriate tests for comparisons.

The discussion is too long and not tightly based on the results, but rather extrapolative. It needs to be adjusted. A clear, evident result is absent. 

The manuscript yields some interest and merit, but needs to be drastically revised.

As a result of careful consideration of the excellent comments of reviewers, the author simplified the setting of the control group to clarify the purpose of the study and improve the reader's understanding. The author's new surgical method is the PMR pTRAM surgery, which removes a part of the rectus abdominis muscle that attaches to the ribcage. Therefore, this study was modified to compare the PMR pTRAM group with the conventional pTRAM (Classical pTRAM) that uses all muscles. Existing studies confuse readers because they subdivided groups including the case where the lateral 1/3 of the rectus abdominis muscle was preserved in the abdomen and the case where it was not, and there were difficulties in statistical analysis. The author made major revisions to the manuscript, especially the discussion section, and performed statistical analysis again.

In particular, I am especially grateful for your comments when revising the discussion section. Significant parts have been deleted and re-written focusing on the results and important issues.
